# Is Synthetic Data all We Need?
# Benchmarking the Robustness of Models Trained with Synthetic Images

Krishnakant Singh[1]     Thanush Navaratnam[*,1]     Jannik Holmer[*,1]     Simone Schaub-Meyer[1,2]
Stefan Roth[1,2]

[1] Department of Computer Science, TU Darmstadt     [2]hessian.AI

## Abstract

*A long-standing challenge in developing machine learning approaches has been the lack of high-quality labeled data. Recently, models trained with purely synthetic data, here termed* synthetic clones, *generated using large-scale pre-trained diffusion models have shown promising results in overcoming this annotation bottleneck. As these synthetic clone models progress, they are likely to be deployed in challenging real-world settings, yet their suitability remains understudied. Our work addresses this gap by providing the first benchmark for three classes of synthetic clone models, namely supervised, self-supervised, and multi-modal ones, across a range of robustness measures. We show that existing synthetic self-supervised and multi-modal clones are comparable to or outperform state-of-the-art real-image baselines for a range of robustness metrics – shape bias, background bias, calibration, etc. However, we also find that synthetic clones are much more susceptible to adversarial and real-world noise than models trained with real data. To address this, we find that combining both real and synthetic data further increases the robustness, and that the choice of prompt used for generating synthetic images plays an important part in the robustness of synthetic clones.*

## 1. Introduction

Most modern machine learning methods are bottlenecked in performance by the quality and quantity of labeled data. Several works [5, 39, 43] have shown that the generalization error of neural networks follows the neural scaling law with respect to the dataset size, *i.e.* the test error reduces linearly with the log of the dataset size. Moreover, the datasets' diversity [53] and fairness [63] are also factors that play an important role in the generalization performance of modern

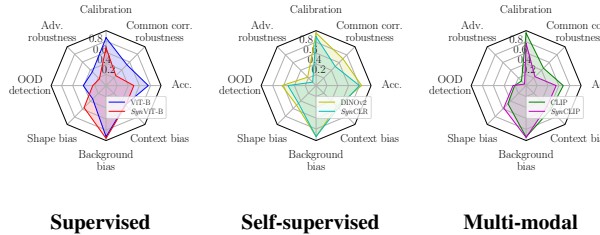

**Supervised**          **Self-supervised**          **Multi-modal**

Figure 1. **Robustness of synthetic clones *vs*. real-image baselines** for different model classes. Self-supervised and multi-modal synthetic clones are close in performance on various robustness measures to baseline models trained on real images. All synthetic clones suffer in performance w.r.t. adversarial and common corruption robustness.

neural networks. Unfortunately, curating diverse, fair, and large datasets is time-consuming and expensive.

The advent of large-scale image generation models like Stable Diffusion [64] has revived the interest in utilizing generated images to train models for various downstream tasks in hopes of alleviating the need for high-quality annotations. Models like [22, 67, 89] use only generated images from Stable Diffusion for supervised training of a downstream classifier. [22, 31, 73] show that it is also possible to train self-supervised models like SimCLR [14] and multimodal models like CLIP [60] using *only* synthetic images and prompts. These models can match or outperform their counterparts trained on real data for downstream tasks like classification and segmentation. We term such models that are trained using only generated data as *synthetic clones*.

Modern machine learning models are increasingly employed in solving real-world problems like autonomous driving and automated medical assistance [9]. With the rapid progress of using synthetic data for training models, it is imperative that we understand how robust these models are before deploying them in the real world. Recent work on synthetic clones [22, 67, 73] has not focused on evaluating the robustness of these models. Yet, models trained

---
Corresponding author: krishnakant.singh@visinf.tu-darmstadt.de

[*]Joint second authors with equal contribution. These authors were responsible for the initial version of common corruption experiments.

on synthetic or generated datasets have been known to suffer from shortcomings such as model collapse [19, 70], *i.e.* when the model forgets the long tail classes or learns a different distribution than the training dataset.

Our work aims to provide a comprehensive benchmark for the robustness of synthetic clone models compared to state-of-the-art (SOTA) baseline models that are trained on real image datasets. We benchmark three classes of synthetic clone models – supervised [22, 67], self-supervised [73], and multi-modal [22] ones – against nine strong baseline models trained using real images. We evaluate robustness metrics regarding shape, background, and context bias. We also benchmark these models against adversarial and common image corruptions. Finally, we test how well these models are calibrated in comparison to models trained with real data. Our results are visually summarized in Fig. 1.

To overcome some of the drawbacks of using synthetic data alone, we conduct extensive ablations regarding how the robustness of synthetic clones changes with *(i)* joint training with synthetic and real data, *(ii)* increasing the number of synthetic samples, and *(iii)* the effect of prompts when generating images with Stable Diffusion.

Let us summarize our findings: *(i)* On many robustness metrics (calibaration, background bias, shape bias, *etc.*) self-supervised and multi-modal models trained on synthetic data perform on par with their counterparts trained on real imagery. *(ii)* Supervised synthetic models, on the other hand, lag behind baselines trained on real datasets w.r.t. several key robustness measures like calibration, OOD detection, adversarial robustness, *etc.* *(iii)* Synthetic clones are much more vulnerable to adversarial and common corruption than models trained with real images. *(iv)* A mixture of real and synthetic data is the best combination for obtaining robustness. *(v)* The choice of prompt for image generation plays a crucial role in the robustness of synthetic clones.

## 2. Related Work

**Self-supervised learning (SSL)** methods [7, 12, 33, 34] have emerged as promising alternatives to solve the data annotation bottleneck. These models learn by solving pretext tasks like context prediction [18], image denoising [79], patch prediction [18], and many others [11, 56, 59, 86, 87]. In recent years, they have come increasingly close to supervised models. For example, the downstream classification accuracy for DINOv2 [57] with supervised linear probing is 84.5% (using ViT-B) while that of EfficientNet [84], a strong supervised model, is 88.4% on the ImageNet-1K dataset [15]. However, SSL methods suffer from scaling issues, *i.e.* augmenting an already large-scale dataset in size has little effect on the model performance [29]. Another approach is using large-scale uncurated multimodal web data [60]. However, this data is often noisy, biased, and limited in diversity (*e.g.*, certain concepts may have only a few data

points [24, 58]).

**Generative neural networks** are a class of models that, given random noise samples, learn to transform these noise samples into data. Modern generative models can broadly be categorized into *implicit models*, such as GANs [3, 10, 27, 45] and diffusion-based models [40, 72], or *explicit models* like normalizing flows [13, 17] and VAEs [48, 77]. Diffusion models are SOTA for image generation since they address the limited diversity and image quality issues, which impaired the use of previous generative models [83].

**Synthetic data** has found usage in a myriad of computer vision tasks or applications like semantic segmentation [61], object detection [65], and autonomous driving [1]. Recently, generated data from large-scale pre-trained diffusion models was used to train better object classification models [4]. Particularly, [35, 67] showed that synthetic data is extremely useful in transfer learning, zero-shot, and few-shot classification. [22, 31, 74] show that even training of large-scale self-supervised models, such as CLIP [60] and SimCLR [14], is possible with synthetic data. Our work focuses on such synthetic clone models where the training data was generated using large-scale pre-trained diffusion models. We use diffusion models because of the superior quality and diversity of the generated data.

**Robustness.** An often overlooked aspect when evaluating models trained with synthetic datasets is evaluating them for robustness. Recently, some efforts have been made to benchmark models trained with synthetic data for adversarial robustness [69] and out-of-distribution (OOD) detection [6]. Still, no comprehensive robustness evaluation of these models exists. In our work, we aim to benchmark synthetic clone models in a more comprehensive manner and on various robustness benchmarks. Besides adversarial robustness [28] and OOD detection, we also benchmark these models on common 2D and 3D image corruptions [36, 44], and w.r.t. shape bias [25], context bias [46], background bias [52], and calibration [30]. Previous works [6, 69] have only benchmarked small-scale supervised synthetic models, while we analyze synthetic clones trained with 100s of millions of synthetic images across three classes of models, namely supervised, self-supervised, and multi-modal ones.

## 3. Background: Synthetic Clones

Before analyzing various synthetic clone models below, let us briefly recapitulate how synthetic images can be generated using diffusion models and how various classes of models have been trained on these synthetic images.

**Synthetic data generation.** The synthetic images in synthetic clone models [22, 31, 67, 73] are typically generated using large-scale pre-trained image generation models, *e.g.*, Stable Diffusion [64] or Imagen [66]. The input to the generation model is Gaussian noise and a conditional text

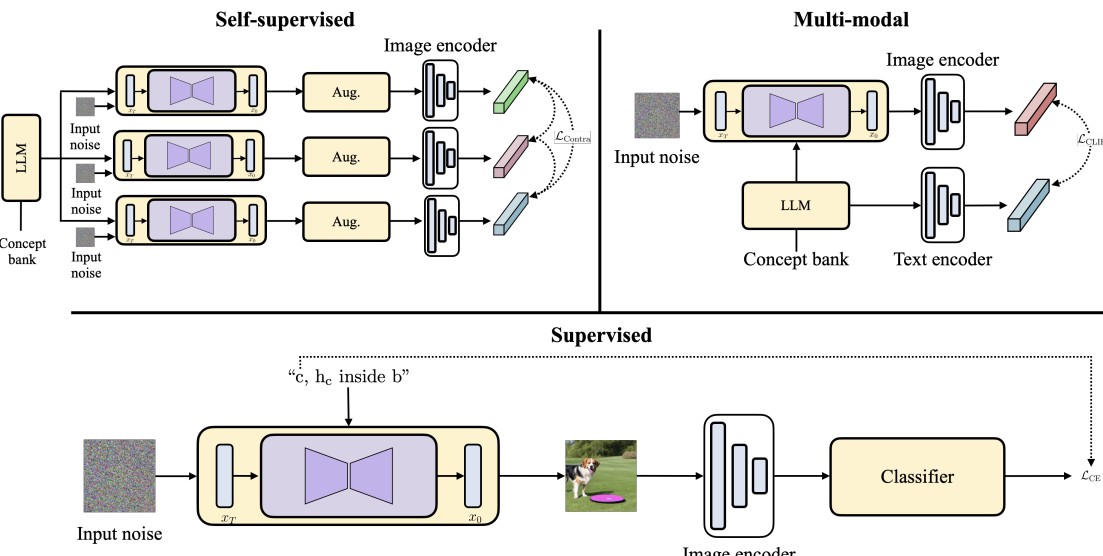

Figure 2. **Setups for training different classes of models using synthetic images.** Supervised learning *(bottom)* uses the ground-truth label for conditionally generating a synthetic image, while self-supervised *(top left)* and multi-modal methods *(top right)* make use of a concept bank along with a large language model (LLM) for prompt generation. Please see text for more details.

prompt. Synthetic clones can be divided into three categories, namely supervised synthetic models, self-supervised synthetic models, and multi-modal synthetic models. We now describe how each model creates the prompt for generating the image and which losses are used to train the model.

**Supervised models using generated data.** For training a supervised classifier, Sarıyıldız et al. [67] first generate an image using Stable Diffusion conditioned on the prompt "c, $h_c$ inside b". Here, c is the ground-truth class name sampled from all class labels of a dataset (*e.g.*, ImageNet-1K [15]), $h_c$ is the hypernym associated with c, and b denotes one of the 365 classes from the Places365 [88] dataset. A hypernym of c is the parent node of c in the WordNet [23] hierarchy. The classifier is then trained end to end with the cross-entropy loss ($\mathcal{L}_{CE}$) between the predicted label of the generated image and the sampled ground-truth class label used for generating the image; see Fig. 2 *(bottom)*. Sarıyıldız et al. [67] created 1.2M such prompts and generated corresponding images to train a ResNet50 model. Similarly, Fan et al. [22] used just class names "c" for generating 16M images. They then train a ViT-B model on the generated images and ground-truth class labels.

**Self-supervised models using generated data.** Synthetic self-supervised models, namely *Syn*CLR [73] and StableRep [74], first sample a concept label from a concept bank. The concept bank is typically constructed using extracted synsets of WordNet [23] or common unigrams, bigrams, and titles from Wikipedia [85]. This sampled concept label is then fed into a large language model (LLM) [2, 42, 76] for generating extra contextual information. The

final prompt is formed by concatenating the concept label and the contextual information. This prompt is then used to generate $n$ images. After this, several augmentations (Aug.) also used in the SimCLR model [14] are applied. The *Syn*CLR model is trained using a multi-positive contrastive loss ($\mathcal{L}_{Contra}$) [47, 73], see in Fig. 2 *(upper left)*.

**Multi-modal model using generated data.** The multi-modal synthetic CLIP [22, 31] models also use a concept label sampled from a concept bank. This concept label, along with a random place label sampled from the classes of Places365 dataset, is fed into an LLM [2] for generating a caption, which is subsequently used for conditional image generation. These images are used to train a CLIP model [60] using a contrastive loss between the generated image and the prompt that was used for generating the image. The architecture is shown in Fig. 2 *(upper right)*.

## 4. Robustness Analysis

**Setup.** We divide the models to be analyzed into supervised, self-supervised, and multi-modal models. For synthetic supervised models, we use a ResNet50 from [67] and a ViT-B model from [22], which were trained on approx. 1M images generated using prompts as described in Sec. 3. The class labels used for creating the prompts were sampled from the classes of the ImageNet-1K dataset [15]. For clarity of notation, we term them *Syn*ResNet50 and *Syn*ViT-B for all our experiments. We compare these models against strong supervised models trained on the real ImageNet-1K dataset like ResNet50 [32], ViT-B [20], DeiT-III [75], Swin transformer [50], and ConvNeXt [51]. All baselines are

from the PyTorch Image Models library [80].

For the self-supervised case, we use the *Syn*CLR model [73], which has been trained on 600M synthetic images. We use SOTA self-supervised models like DINOv2 [57], MAE [34], and MOCOv3 [33] trained on ImageNet-1K as self-supervised baselines. All checkpoints for the baseline models were obtained from the timm library. For a fair comparison, we use the ViT-B [20] backbone with a patch size of 16 for all models. We perform linear probing on all self-supervised models, training a single-layer linear classification head on the top of these models for 90 epochs using the ImageNet-1k [15] dataset. We searched over ten learning rates to find the optimal linear classifier for each model.

Finally, for the multi-modal case, we analyze the synthetic CLIP model from [22], which we term as *Syn*CLIP, trained on 371M synthetic images. We compare this model with the CLIP implementation from OpenCLIP [41], trained on 400M real images. We used the ViT-B backbone for these models to allow for a fair comparison. For CLIP and *Syn*CLIP we report the zero-shot results.

## 4.1. Calibration

As neural networks become adopted for safety-critical tasks like autonomous driving and healthcare, it is not only important to predict accurate results, but also to accurately report the confidence in their prediction [30]. Calibration can help to understand how reliable the model's prediction is and whether an end user can trust the model's output. The calibration of neural networks is commonly measured using the Expected Calibration Error (ECE) [55]. The ECE measures the expected absolute difference between the model confidence and the model accuracy. In our work, we study the effect that training on synthetic images has on the calibration of a model compared to training with real data.

We report the results for the ECE metric with 20 bins for all models. Fig. 3 shows the results for both in-distribution (ID) calibration (train and test splits are from the same dataset) on the ImageNet-1k dataset [15] and for out-of-distribution (OOD) calibration (train and test split are from different datasets) on the ImageNet-R [37] and ImageNet-A [38] datasets. We can conclude the following:

> **Observation 1:** *Synthetic clones are mostly well calibrated for the in-distribution case and even to some extent out-of-distribution on ImageNet-R. The OOD calibration of synthetic clones suffers on ImageNet-A.*

This may be because the synthetic data generated from pre-trained diffusion models (trained on data scraped from the web) already captures the distribution of the ImageNet (images scraped from the web) and ImageNet-R (consisting of cartoons and sketches, which are abundant on the internet) datasets. ImageNet-A, on the other hand, consists of

naturally adversarial examples that are hard to find on the internet; hence, synthetic clones and even baseline models trained on real images exhibit a rather poor calibration for this dataset. However, models trained with real datasets are generally better calibrated for ImageNet-A, likely due to the inherent noise in the dataset (see also Sec. 4.2).

**Out of distribution (OOD) detection**  deals with finding out how well a model can distinguish between samples from the training data distribution (ID – in distribution) and samples from another distribution. OOD detection is critical to increasing an end users' trust in the safety and reliability of the model. We thus aim to evaluate how training on synthetic data affects a model's capability for OOD detection.

The OOD detection task can be formulated as a binary classification task on the model's predictive probability. A model $F$ with weights $\theta$ classifies an input sample $x_i$ as ID if the maximum predictive probability of the sample is higher than a pre-defined threshold value $\tau$, *i.e.* $\max F_\theta(x_i) \geq \tau$, and as OOD if $\max F_\theta(x_i) < \tau$. OOD detection can be evaluated using standard metrics for binary classification, such as the area under the receiver operating characteristic curve (AUROC). We also report the false positive rate of OOD samples when the true positive rate of in-distribution samples is at 95% (FPR@95). Tab. 1 shows the results of all models on three OOD datasets, namely SUN397 [81], Places365 [88], and iNaturalist [78], where ImageNet-1K is the ID dataset. We conclude the following:

> **Observation 2:** Syn*CLR and* Syn*CLIP are comparable to the baseline models in their category for OOD detection. Even with 16 times more data than the baseline,* Syn*ViT-B clearly lags behind supervised models trained with real data.*

## 4.2. Robustness

**Adversarial robustness.**  Adversarial learning aims to understand model robustness to examples manipulated by an adversary in a way that the examples seem similar to the human eye but change the model's predictions. In our work, we want to explore whether models trained on synthetic data are more susceptible to adversarial attacks. We use two popular white-box attacks, the Fast Gradient Sign Method (FGSM) [28] and the Projected Gradient Descent (PGD) attack [52]. These white-box attacks require that the model's gradient be known to the adversary. The FGSM attack perturbs the input image with the gradient of the model's prediction w.r.t. its input, scaled by a small step $\epsilon$. This can be written as $\hat{x}_i = x_i + \epsilon \nabla_{x_i} J(\theta, x_i, y_i)$, where $x_i$ denotes the input image, $\nabla_{x_i} J$ denotes the gradient of the loss function w.r.t. $x_i$, and $y_i$ denotes the label for the input image $x_i$.

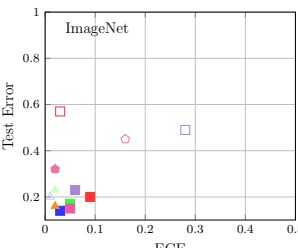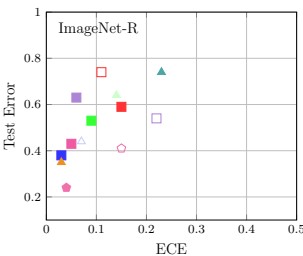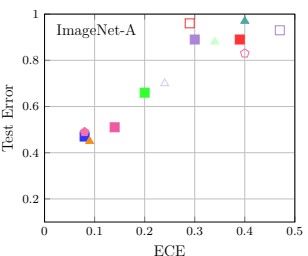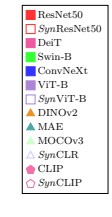

Figure 3. **Test error *vs*. ECE for ID and OOD datasets.** We report the resulting ECE metric and test error metrics for both ID (ImageNet) and OOD datasets (ImagNet-{R,A}). Filled markers indicate real models, empty markers indicate synthetic clones.

Table 1. **OOD detection with ImageNet-1K being in-domain (ID).** We report the AUROC and FPR@95 metrics for the OOD detection task on the three OOD datasets, namely SUN, iNatualist, and Places. In addition, we also report the avg. performance of all models on all three datasets. The best performing model in each category is highlighted in **bold**.

| Model | No. of training images | SUN AUROC (↑) | SUN FPR@95 (↓) | iNaturalist AUROC (↑) | iNaturalist FPR@95 (↓) | Places AUROC (↑) | Places FPR@95 (↓) | Avg. AUROC (↑) | Avg. FPR@95 (↓) |
|---|---|---|---|---|---|---|---|---|---|
| ResNet50 | 1.2M | **0.84** | 0.65 | **0.91** | 0.47 | **0.82** | 0.69 | **0.86** | 0.60 |
| *Syn*ResNet50 | 1.2M | 0.70 | 0.83 | 0.72 | 0.89 | 0.67 | 0.87 | 0.70 | 0.86 |
| Swin-B | 1.2M | 0.79 | **0.63** | 0.86 | **0.45** | 0.77 | 0.69 | 0.81 | **0.59** |
| ConvNeXt | 1.2M | 0.76 | 0.68 | 0.89 | **0.45** | 0.74 | 0.71 | 0.80 | 0.61 |
| DeiT | 1.2M | 0.80 | 0.66 | 0.89 | 0.48 | 0.80 | 0.68 | 0.83 | 0.61 |
| ViT-B | 1.2M | 0.81 | 0.64 | 0.90 | **0.45** | 0.80 | **0.67** | 0.84 | **0.59** |
| *Syn*ViT-B | 16M | 0.76 | 0.74 | 0.75 | 0.75 | 0.72 | 0.79 | 0.74 | 0.76 |
| MAE | 1.2M | 0.76 | 0.84 | 0.87 | 0.71 | 0.75 | 0.86 | 0.79 | 0.80 |
| DINOv2 | 142M | **0.88** | **0.49** | **0.98** | **0.09** | **0.87** | **0.53** | **0.91** | **0.37** |
| MOCOv3 | 1.2M | 0.84 | 0.65 | 0.94 | 0.35 | 0.84 | 0.66 | 0.87 | 0.55 |
| *Syn*CLR | 600M | 0.85 | 0.58 | 0.95 | 0.24 | 0.83 | 0.63 | 0.88 | 0.48 |
| CLIP | 400M | **0.82** | **0.74** | 0.68 | 0.88 | **0.78** | **0.76** | **0.76** | 0.79 |
| *Syn*CLIP | 371M | 0.73 | 0.75 | **0.74** | **0.75** | 0.70 | 0.79 | 0.72 | **0.76** |

The PGD attack is an iterative version of the FGSM attack, followed by a projection of the adversarial input to an $\epsilon$ ball around the input $x$. The $\epsilon$ value denotes the maximum perturbation allowed. We use $\epsilon$ values of 1/255 for the PGD and FGSM attacks. The number of steps is set to 20 for the PGD attack. We report the accuracy of the clean and the adversarial examples from the test set. We define the adversarial robustness metric, $R_{\text{adv}}$, as the relative accuracy between adversarial and clean samples as $R_{\text{adv}} = \frac{\text{Acc}_{\text{adv}}}{\text{Acc}_{\text{clean}}}$, where $\text{Acc}_{\text{adv}}$ is the accuracy on the adversarial samples and the $\text{Acc}_{\text{clean}}$ is the accuracy on the clean samples. Tab. 2 shows the results. We can conclude the following:

> **Observation 3:** *Synthetic clone models are significantly more vulnerable to adversarial examples, particularly supervised synthetic clones, than models trained with real data. The self-supervised synthetic clone model trained with large amounts of synthetic data, i.e. SynCLR, is loosely comparable to real-image baseline models in its respective category.*

We find that MAE [34] performs the worst among all models (synthetic and real) regarding adversarial robustness, indicating that the training objective along with the training dataset size are important factors in determining a model's adversarial robustness.

Table 2. **Adversarial robustness results** (in %). We report the clean and adversarial accuracy. Also, we report the relative adversarial robustness ($R_{\text{adv}}$) metric for each model.

| Model | $\text{Acc}_{\text{clean}}$(↑) | FGSM $\text{Acc}_{\text{adv}}$(↑) | FGSM $R_{\text{adv}}$(↑) | PGD $\text{Acc}_{\text{adv}}$(↑) | PGD $R_{\text{adv}}$(↑) |
|---|---|---|---|---|---|
| ResNet50 | 80.12 | 26.95 | 33.64 | 16.71 | 20.85 |
| *Syn*ResNet50 | 42.89 | 2.12 | 4.95 | 1.27 | 2.96 |
| Swin-B | 83.08 | 48.59 | 58.49 | 23.71 | 28.54 |
| ConvNext | **85.52** | 42.19 | 49.33 | 17.51 | 20.47 |
| DeiT | 84.59 | **53.22** | **62.92** | **35.51** | **41.98** |
| ViT-B | 76.78 | 27.39 | 35.67 | 20.45 | 26.64 |
| *Syn*ViT-B | 50.96 | 8.84 | 17.35 | 5.06 | 9.92 |
| MAE | 67.59 | 0.67 | 0.99 | 1.34 | 1.98 |
| DINOv2 | **84.49** | **19.10** | **22.61** | **18.71** | **22.14** |
| MOCOv3 | 76.66 | 13.21 | 17.23 | 9.11 | 11.88 |
| *Syn*CLR | 80.46 | 7.31 | 9.08 | 6.18 | 7.68 |
| CLIP | **68.27** | **8.75** | **12.82** | **6.31** | **9.24** |
| *Syn*CLIP | 55.11 | 2.40 | 4.35 | 2.02 | 3.67 |

**Robustness against common corruptions.** Next, we evaluate the performance of all models on real-world noise corruptions that occur frequently. For this, we evaluate on the ImageNet-C [36] and ImageNet-3DCC [44] datasets. ImageNet-C consists of 19 naturally occurring image corruptions like Gaussian noise, shot noise, motion blur, elastic transforms, *etc*. ImageNet-3DCC includes 12 common corruptions that take depth into account, *e.g.*, $z$-axis blur,

Table 3. **Common corruptions robustness results** (in %). We report the individual and average accuracy for various 2D and 3D common corruptions. We also report the relative drop in average accuracy (Avg. $R_{cc}$) for all models.

| Noise | ResNet50 | SynResNet50 | Swin-B | ConvNeXt | DeiT | ViT-B | SynViT-B | MAE | DINOv2 | MOCOv3 | SynCLR | CLIP | SynCLIP |
|---|---|---|---|---|---|---|---|---|---|---|---|---|---|
| Acc$_{Clean}$ | 80.12 | 42.89 | 83.08 | **85.52** | 84.59 | 76.78 | 50.96 | 67.59 | **84.49** | 76.66 | 80.46 | **68.27** | 55.11 |
| Shot noise | 56.31 | 5.11 | 66.33 | 72.31 | **75.41** | 57.95 | 29.03 | 34.47 | **73.45** | 57.14 | 43.23 | **44.13** | 19.85 |
| Motion blur | 47.55 | 5.56 | 59.87 | 67.30 | **70.85** | 49.58 | 19.91 | 27.70 | **67.24** | 48.68 | 37.47 | **38.63** | 16.67 |
| Snow | 44.32 | 7.07 | 57.62 | 65.38 | **69.02** | 45.71 | 18.33 | 25.87 | **67.17** | 46.27 | 40.38 | **38.22** | 16.64 |
| Pixelate | 45.32 | 7.38 | 58.11 | 66.07 | **69.66** | 46.95 | 19.96 | 27.89 | **69.06** | 48.47 | 41.33 | **40.14** | 17.26 |
| JPEG compression | 47.03 | 7.09 | 58.92 | 67.51 | **70.37** | 49.82 | 20.91 | 30.38 | **70.70** | 51.60 | 39.66 | **41.81** | 16.34 |
| Near focus | 64.55 | 27.28 | 69.11 | 75.23 | **75.82** | 63.17 | 33.98 | 47.18 | **77.33** | 65.14 | 67.57 | **56.90** | 34.64 |
| Far focus | 60.94 | 24.76 | 65.93 | 72.53 | **73.28** | 60.13 | 31.34 | 43.96 | **75.35** | 61.77 | 63.85 | **53.89** | 32.04 |
| Fog 3D | 58.80 | 23.22 | 63.67 | 70.17 | **71.00** | 58.11 | 30.57 | 40.17 | **72.87** | 58.64 | 60.48 | **51.29** | 30.88 |
| XY motion blur | 54.30 | 19.42 | 60.12 | 66.86 | **67.94** | 53.95 | 26.71 | 35.25 | **69.11** | 54.43 | 54.78 | **47.20** | 27.05 |
| Z motion blur | 50.43 | 16.77 | 56.85 | 64.28 | **65.52** | 50.12 | 22.97 | 32.07 | **66.59** | 50.65 | 50.10 | **43.90** | 23.64 |
| Avg. Acc$_{cc}$ ($\uparrow$) | 52.96 | 14.37 | 61.65 | 68.76 | **70.89** | 53.55 | 25.37 | 34.50 | **70.89** | 54.28 | 49.88 | **45.61** | 23.50 |
| Avg. $R_{cc}$ ($\uparrow$) | 66.10 | 33.50 | 74.21 | 80.40 | **83.80** | 69.74 | 49.78 | 51.04 | **83.90** | 70.81 | 61.99 | **66.81** | 42.64 |

far and near focus errors, *etc*. Due to time and resource constraints, we report the results only on ten common corruption tasks (five each from ImageNet-C and ImageNet-3DCC). We report the accuracy of the clean and corrupted samples and the average accuracy over all corruptions. We also report the Avg. $R_{cc}$ metric, which is defined as the relative accuracy between the clean samples and the average accuracy over all corruptions, *i.e.* Avg. $R_{cc} = \frac{\text{Avg. Acc}_{cc}}{\text{Acc}_{clean}}$. The results are given in Tab. 3 and yield the following conclusion:

> **Observation 4:** *Synthetic clones are significantly less robust to common corruptions in images than baselines trained with real images.*

The Avg. $R_{cc}$ is significantly lower for synthetic clones across all categories of models. Real datasets inherently have these common corruptions present in the imagery, hence training on real data already makes the resulting models more robust to noise. Synthetic images currently lack these corruptions, making synthetic clones highly susceptible to common image corruptions.

## 4.3. Biases

**Context bias.** We define context bias as the affinity of a model to use contextual cues, *e.g.*, location for classifying objects, rather than actually using the object appearance. This context bias exists because most large-scale datasets consist of uncurated data scraped from the internet. For example, images of airplanes in a forest are highly unlikely when compared to airplanes on a taxiway. We use the FOCUS (Familiar Objects in Common and Uncommon Settings) dataset [46] to evaluate the context bias, which consists of around 21K images. Each image in the dataset is annotated with the object class, the time of day, location, and weather labels. FOCUS subdivides the dataset into a subset of common and uncommon samples. Uncommon samples are uncommon in the real world, like "airplane in forest"

Table 4. **Context bias results** (in %). CB$_k$ denotes the context bias with $k$ uncommon attributes.

| Model | CB$_1$ ($\uparrow$) | CB$_2$ ($\uparrow$) |
|---|---|---|
| ResNet50 | 61.27 | 38.70 |
| SynResNet50 | 62.33 | 44.49 |
| Swin-B | 68.83 | 54.85 |
| ConvNext | **70.20** | **55.57** |
| DeiT | 68.19 | 55.19 |
| ViT-B | 67.21 | 49.71 |
| SynViT-B | 66.13 | 50.29 |
| MAE | 59.75 | 46.53 |
| DINOv2 | 69.11 | 54.46 |
| MOCOv3 | 62.29 | 44.95 |
| SynCLR | **70.04** | **58.42** |
| CLIP | **76.77** | **63.09** |
| SynCLIP | 71.47 | 54.39 |

or uncommon in the ImageNet dataset due to labels used for its construction (*e.g.*, there is no label for seaplane in ImageNet). The dataset is partitioned into mutually exclusive partitions $P_k$ where $k$ is the number of uncommon attributes. The total dataset is divided into four partitions, $P_0$ (containing only common objects) to $P_3$ (containing three uncommon attributes). We report the CB$_k$ metrics (Context Bias with $k$ uncommon attributes), which is defined as the relative accuracy between the accuracy on the partition with no uncommon attributes $P_0$ and a partition with $k$ uncommon attributes $P_k$, *i.e.* CB$_k = \frac{\text{Acc}_{P_k}}{\text{Acc}_{P_0}}$. For example, CB$_2$ measures the relative accuracy between $P_0$ and $P_2$. The results are given in Tab. 4 and yield the following:

> **Observation 5:** *Self-supervised synthetic clones are robust to changes in context compared to baseline supervised and self-supervised models trained with real data. The supervised synthetic clone SynViT-B is comparable in performance to the ViT-B model trained on real data. Meanwhile, SynCLIP is more prone to changes in context compared to CLIP, but it is still comparable to models like DINOv2 and ConvNeXt.*

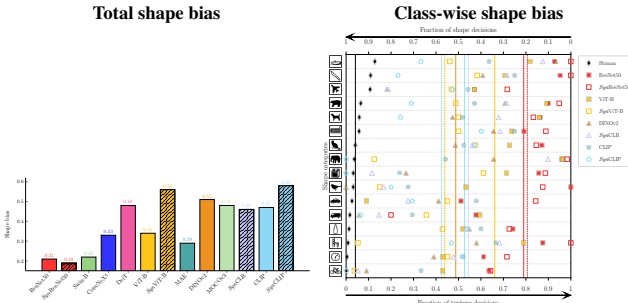

**Figure 4. Shape bias.** *(left)* Average shape bias of models trained with synthetic (dashed bars) and real data. Synthetic clones are generally more biased toward shape than texture compared to models trained with real datasets. *(right)* Class-wise shape bias of synthetic clones and their counterparts trained using real data. Solid and dashed lines represent the mean shape bias of models trained with real images and synthetic images, respectively.

**Shape-texture bias.** Children learn to recognize and organize objects based on shape and are more biased towards object shape rather than color and textures [16, 71]. It has been shown [25, 54] that biasing a network towards shape increases its robustness against common corruptions. This suggests that the robustness of neural networks generally benefits from biasing towards categorizing objects by shape rather than textures. Generated images from GANs typically have high-frequency artifacts (indicating high texture bias) [21, 68]. Diffusion models also exhibit similar patterns, though these are more muted [62]. Such artifacts contrast with real images that do not contain these high-frequency artifacts. To understand if training on synthetic images from Stable Diffusion biases the networks towards texture, we use the cue conflict dataset [25]. This dataset consists of about 1200 images of 16 classes where the texture and shape of an image are in conflict with each other. Fig. 4 shows the shape bias of all the models averaged across all classes. We also show the class-wise shape bias results for synthetic clones and some baseline models. We conclude the following:

> **Observation 6:** *Synthetic clones tend to be more shape-biased than texture-biased.* In particular, *Syn*CLIP outperforms all models on the shape bias metric, while *Syn*ViT-B outperforms all classification and self-supervised models. The *Syn*CLR model has comparable performance to the MOCOv3 model and outperforms the MAE model on the shape-bias metric.

A similar result was observed in [8] with synthetic data from StyleGANv2 [45] models. Our results indicate that synthetic data is diverse in terms of shape, leading to a higher shape bias of synthetic clone models, but this could indicate that the generated images lack texture diversity,

Table 5. **Background bias results** (in %). BG-Gap metric reports the drop in performance by just changing the background to a different class than the foreground class.

| Model | Original Acc. (IN-9L, ↑) | Mix-Same Acc. (↑) | Mix-Rand Acc. (↑) | BG-Gap (↓) |
|---|---|---|---|---|
| ResNet50 | 95.43 | 87.04 | 81.36 | 5.68 |
| *Syn*ResNet50 | 66.44 | 44.35 | 35.83 | 8.52 |
| Swin-B | 96.57 | 88.32 | 82.57 | 5.75 |
| ConvNeXt | **97.98** | **93.95** | **90.40** | 3.56 |
| DeiT | 97.70 | 93.28 | 89.98 | **3.31** |
| ViT-B | 95.98 | 87.53 | 79.63 | 7.90 |
| *Syn*ViT-B | 87.70 | 77.01 | 71.60 | 5.41 |
| MAE | 57.36 | 46.47 | 40.57 | **5.90** |
| DINOv2 | **97.95** | **91.93** | **85.95** | 5.98 |
| MOCOv3 | 95.01 | 83.63 | 74.17 | 9.46 |
| *Syn*CLR | 96.22 | 86.59 | 80.37 | 6.22 |
| CLIP | **93.31** | **83.09** | **77.19** | 5.90 |
| *Syn*CLIP | 84.79 | 71.16 | 65.83 | **5.33** |

making the network rely more on shape for classification.

**Background bias.** The background bias of models can be used to identify if the model is using the background of the image to make the classification decision instead of using the object itself. Learning if a model is biased towards the background is an effective way to understand if the model has learned shortcuts [26] instead of learning good features for the given category. For evaluating a model's background bias, we utilize the Mixed-Rand and Mixed-Same partitions from the IN-9L dataset [82]. The Mixed-Rand dataset segments the foreground object in an image and switches the original background with a random background from a different class label, while the Mixed-Same partition places the segmented foreground object on a random background from the same class label. Tab. 5 shows the accuracy of all models on the original, Mixed-Rand, and Mixed-Same partitions from the IN-9L dataset, along with BG-Gap. The BG-Gap measures the difference in performance between accuracies on the Mixed-Rand and Mixed-Same datasets and assesses how decisions can be manipulated just by changing the background to a different class than the foreground. We conclude the following:

> **Observation 7:** *Synthetic clones perform on par in terms of background bias with a SOTA baseline model trained with real data.*

In general, we found all models (synthetic and real) to be very robust to background changes.

### 4.4. Ablations

We now look at three important factors that affect the robustness of synthetic clone models. We use the models from [22] for these ablations (including all the CLIP models).

**Effect of prompts.** Here, we analyze the effect that the prompt has on the robustness of the synthetic clone models.

Table 6. **Effect of prompt type and dataset size on various performance metrics for the supervised *Syn*ViT-B model.** $R_{adv}$ (FGSM) denotes the adversarial robustness for the FGSM attack, and $R_{cc}$ (2DCC) the robustness for 2D common corruptions. **Bold** indicates the best performance within a prompt type, and color indicates the best performance across all prompts and dataset sizes.

| Metric | Class name | | | | Captions | | | | CLIP templates | | | |
|---|---|---|---|---|---|---|---|---|---|---|---|---|
| | 1M | 4M | 8M | 16M | 1M | 4M | 8M | 16M | 1M | 4M | 8M | 16M |
| Acc. (↑) | 0.44 | 0.49 | 0.50 | **0.51** | 0.50 | 0.58 | 0.59 | **0.60** | 0.45 | 0.53 | 0.54 | **0.55** |
| $R_{adv}$ (FGSM, ↑) | 0.14 | **0.18** | 0.17 | 0.17 | 0.12 | 0.16 | 0.16 | **0.17** | 0.15 | 0.17 | **0.18** | **0.18** |
| $R_{cc}$ (2D-CC, ↑) | 0.39 | 0.42 | **0.43** | 0.42 | 0.37 | 0.43 | **0.44** | **0.44** | 0.44 | 0.51 | **0.54** | **0.54** |
| $CB_2$ (↑) | 0.47 | 0.47 | 0.47 | **0.50** | 0.52 | 0.58 | 0.58 | **0.59** | 0.47 | **0.53** | 0.50 | 0.50 |
| Shape Bias (↑) | 0.39 | 0.55 | **0.56** | **0.56** | 0.33 | 0.42 | **0.47** | 0.45 | 0.57 | **0.71** | **0.71** | 0.69 |
| BG-Gap (↓) | 0.71 | 0.66 | 0.61 | **0.54** | 0.85 | 0.54 | 0.53 | **0.52** | 0.63 | 0.42 | **0.37** | 0.46 |
| FPR@95 (SUN, ↓) | 0.81 | **0.74** | 0.75 | **0.74** | 0.77 | 0.74 | **0.73** | 0.74 | 0.84 | 0.82 | 0.81 | **0.80** |
| ECE (↓) | 0.33 | 0.31 | 0.29 | **0.28** | 0.25 | 0.18 | 0.17 | **0.16** | 0.30 | 0.24 | **0.23** | **0.23** |

Table 7. **Effect of dataset composition and size on various performance metrics for the *Syn*CLIP model.** **Bold** indicates the best performance within a prompt type, and color indicates the best performance across all dataset compositions and sizes.

| Metric | Real | | | | Synthetic | | | | Synthetic + Real | | | |
|---|---|---|---|---|---|---|---|---|---|---|---|---|
| | 64M | 128M | 256M | 371M | 64M | 128M | 256M | 371M | 64M | 128M | 256M | 371M |
| Acc. (↑) | 0.55 | 0.60 | 0.65 | **0.66** | 0.47 | 0.51 | 0.54 | **0.55** | 0.56 | 0.62 | 0.65 | **0.66** |
| $R_{adv}$ (FGSM, ↑) | 0.09 | 0.07 | 0.10 | **0.12** | **0.05** | 0.03 | 0.04 | 0.04 | 0.09 | 0.08 | 0.10 | **0.12** |
| $R_{cc}$ (2D-CC, ↑) | 0.44 | 0.46 | 0.51 | **0.52** | 0.29 | 0.28 | **0.31** | **0.31** | 0.46 | 0.48 | **0.52** | **0.52** |
| $CB_2$ (↑) | 0.52 | 0.52 | 0.57 | **0.61** | **0.58** | 0.53 | 0.55 | 0.54 | 0.61 | 0.58 | **0.63** | 0.61 |
| Shape Bias (↑) | 0.51 | 0.51 | 0.51 | **0.52** | 0.54 | 0.55 | **0.59** | 0.58 | 0.54 | 0.51 | 0.56 | **0.60** |
| BG-Gap (↓) | 0.73 | 0.76 | **0.57** | 0.72 | 0.65 | **0.51** | 0.57 | 0.53 | 0.73 | 0.63 | 0.63 | **0.56** |
| FPR@95 (SUN, ↓) | 0.92 | 0.84 | **0.81** | 0.82 | 0.86 | **0.75** | **0.75** | **0.75** | 0.86 | 0.81 | 0.82 | **0.78** |
| ECE (↓) | 0.22 | 0.19 | 0.16 | **0.14** | 0.25 | 0.20 | 0.17 | **0.16** | 0.16 | 0.13 | **0.11** | **0.11** |

Tab. 6 shows results for a *Syn*ViT-B model [22] trained on synthetic images generated using different prompts such as *(i)* class names, *(ii)* 80 CLIP templates, *e.g.* "high-quality photo of {class name}", used in evaluating the zero-shot classification performance of the CLIP model, and *(iii)* class names combined with a generated caption from BLIP2 [49], *e.g.* "Tench [class label], a man holding a fish". As can be seen in Tab. 6, captions and CLIP templates are much better for creating robust synthetic clones compared to just using class names. This can be attributed to more diverse images being generated with more descriptive text.

**Effect of adding real data.** Next, we study the effect of using a mixture of real and synthetic image data on the robustness of the CLIP model. Fan et al. [22] trained the CLIP model with a fixed dataset size (for example, 371M images) where the real and synthetic images are picked randomly to create a subset containing both real and synthetic images, which are then used for training the CLIP model. Tab. 7 shows that adding real data as suggested by [22] improves the performance on many key metrics (ECE, adversarial accuracy, shape bias) while remaining comparable on others. Also, we see that training with just synthetic images or a combination of synthetic and real images creates more robust models compared to models trained just on real data.

**Size of generated data.** We evaluate the effect of dataset size on the training of synthetic clones. As seen in Tabs. 6 and 7, adding more data, in general, helps with the robustness of both *Syn*ViT-B and *Syn*CLIP models. In some cases, adding more data may slightly decrease performance, which

can be due to less dataset diversity with increasing dataset size and overfitting of the model with less diverse data.

## 5. Conclusion

Our work is the first to perform a detailed analysis of models trained with synthetic data across different robustness measures. Specifically, we show that certain synthetic clones, namely *Syn*CLIP and *Syn*CLR, perform within tolerable limits of their counterparts trained on real images; this holds for all robustness metrics except for common corruptions and OOD detection. Supervised models, namely *Syn*ViT-B, on the other hand, are outperformed by their real-image counterparts on all metrics except shape bias, which clearly shows the need for better supervised synthetic clones. Through detailed ablations, we find that using captions or CLIP templates produces more robust synthetic clones. Importantly, we find that mixing real data with synthetic data can improve the robustness measures across most metrics. We hope our work encourages the development of more robust synthetic clones.

**Acknowledgements.** This project has received funding from the European Research Council (ERC) under the European Union's Horizon 2020 program (grant agreement No. 866008). The project was also supported in part by the State of Hesse through the project "The Third Wave of Artificial Intelligence (3AI)".

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
