# OpenReview forum: "Is Synthetic Data all We Need? Benchmarking the Robustness of Models Trained with Synthetic Images"
_thecvf.com/CVPR/2024/Workshop/SyntaGen — SyntaGen 2024_

### Official Review · Reviewer_y2Fa · 2024-03-28
**Good paper with clear findings but minor issues**

**Rating:** 6
**Confidence:** 3

**Review:**

This is a good quality paper with aims and objectives clearly written and findings clearly stated. I think it lacks originality as it just conducts robustness measures to models trained in other works. However, I believe it has good-quality findings of enough significance to warrant acceptance. I have provided a summary of the paper from my understanding, as well as a list of specific strengths and weaknesses along with some specific comments.

**Paper Summary**:

This work compares and analyses the robustness of models trained with real images to models trained with synthetic images. They do so across three training settings, supervised, self-supervised and multi-modal. Their analysis reveals that training with synthetic data can achieve equally robust models compared to training on real data for self-supervised and multi-modal models, however, supervised models trained on real data are generally more robust than their synthetically trained counterparts. Additionally, they find that synthetic clones are more vulnerable to adversarial attacks and common corruptions, most likely due to the lack of realistic noise present in synthetic datasets. Synthetic clones also display a stronger shape bias compared to models trained on real images. Lastly, synthetic clones benefit from more training samples, a mixture of real and synthetic images and more diversity in the synthetic training data as controlled by the input prompts during data generation.

**Paper Strengths**:
- Detailed analysis of the robustness of models trained on real images and synthetic clones across a range of tests.
- Clearly written with clear observations at the conclusion of each individual analysis.
- Overall provides an interesting insight into the differences of models trained with real and synthetic images.

**Paper Weaknesses**:
- Low number of synthetic clones relative to real models in results, 2/7 for supervised, 1/4 for self-supervised. This makes it more difficult to spot a common trend among the synthetic clones and therefore draw overall conclusions. Ideally, you would have 1 to 1 for each real model and synthetic clone, i.e. Swin-B and SynSwin-B.

**Additional Comments**:
- "StableDiffusion" in intro should be 2 words i.e. "Stable Diffusion"
- Some tables and figures need more informative captions (figure 3 and tables 1, 2, 3, 4, 5)
- reference 41 has "If you use this software, please cite it as below."

---

### Official Review · Reviewer_nsJz · 2024-04-02
**Review of CVPR 2024 Workshop SyntaGen Submission18**

**Rating:** 8
**Confidence:** 4

**Review:**

This paper provides a comprehensive examination of the robustness of synthetic clone models across various metrics.

Quality: The research is of high quality, with rigorous experiments and a thorough analysis of the results.

Clarity: This paper is well-written and easy-to-follow.

Originality: While there are existing works (e.g., [1]) focusing on benchmarking the robustness of synthetic data, this research proposes some new perpectives and their own observations.

Significance: This work offers insights that could influence future research directions regarding the robustness of synthetic data.

References:
[1] Benchmarking Robustness to Text-Guided Corruptions, CVPRW 2023

---

### Official Review · Reviewer_K16q · 2024-04-04
**Paper provide a comprehensive analysis and benchmark of synthetic-based models for 3 types of classes: self-supervised, supervised, and multi-modal ones. It also gives insights about the advantages and disadvantages of synthetic data versus real data and emphasizes the importance of robustness measures for those models trained on synthetic data.**

**Rating:** 9
**Confidence:** 4

**Review:**

The exposition is well-writen. The paper's novelty lies in its comprehensive evaluation of synthetic-based models, a critical contribution to benchmarking and advancing recent developments in the field.

Pros:

- Give substantial effort for detailed analysis of robustness evaluation: out-of-distribution detection, adversarial robustness & common corruption in images, and shape biases.
- Provide useful insights for each measure which are very informative for broad audiences.
- The finding and contribution of this work is critical for the advancement of using synthetic data for training model.
- A complement to figure 1: neat and comprehensive but its size is a bit squeezing.

Cons:

- IMO, this paper covers almost all essential aspects of training a model with synthetic data and presents a benchmarking protocol for foundational models. Currently, I could not identify any significant shortcomings in the work.
- Just a minor concern: though MAE has the worst performance in adversarial robustness measure (in Tab 2), its results appear to be somewhat ambiguous presented.
- Another minor: Effect of adding real data: it needs to show the precise portion of real data and generated data to better interpret model behavior.

Overall, I recommend a strong acceptance.

---

### Decision · Program_Chairs · 2024-04-06

**Decision:**

Accept

**Comment:**

The paper receives three reviews with positive comments. The reviewers unanimously agree that this is a good quality paper with the comprehensive analysis and benchmark presented in the paper that compares models trained on real and synthetic data, respectively. The paper is a timely contribution and the observations can be of great interest to a broader audience, and therefore the organizers are happy to accept the paper to the workshop.